# Epidemiology and risk factors of Japanese encephalitis in Taiwan, 2010–2022

**Jen-Yu Hsu**[1,2,3]*, **Chien-Ching Hung**[4,5,6], **Tsung-Pei Tsou**[2], **Wan-Chin Chen**[2]

**1** Department of Occupational Medicine and Clinical Toxicology, Taipei Veterans General Hospital, Taipei, Taiwan, **2** Centers for Disease Control, Ministry of Health and Welfare, Taipei, Taiwan, **3** School of Medicine, College of Medicine, National Yang Ming Chiao Tung University, Taipei, Taiwan, **4** Department of Tropical Medicine and Parasitology, National Taiwan University College of Medicine, Taipei, Taiwan, **5** Department of Internal Medicine, National Taiwan University Hospital and National Taiwan University College of Medicine, Taipei, Taiwan, **6** Department of Internal Medicine, National Taiwan University Hospital Yunlin Branch, Yunlin, Taiwan

* b94401012@ntu.edu.tw

## Abstract

### Introduction

Taiwan introduced a two-dose inactivated Japanese encephalitis (JE) mouse brain-derived (JE-MB) vaccine into routine childhood immunization in 1968, with booster vaccination implemented in 1974 and 1983. In 2017, JE-MB vaccine was replaced by a two-dose live-attenuated chimeric vaccine (JE-CV). After implementation of JE vaccination programs, JE cases have shifted from children to adults. In this study, we described the JE epidemiology and identify high-risk groups to further inform vaccine policy.

### Methodology/Principal findings

We extracted data from Taiwan's notifiable disease surveillance database, vital statistics, and employment statistics from 2010 to 2022. Diagnosis of JE was confirmed by JE sero-conversion, a four-fold increase in virus-specific antibodies, a positive JE viral nucleic-acid test, or JE virus isolation. From 2010 to 2022, a total of 313 cases of JE were diagnosed, resulting in an overall incidence rate of 0.10 cases per 100,000 person-years and a mortality rate of 0.006 per 100,000 population per year. Among these patients, 64% were male, and the median age was 51 years (range 0–82). Compared with people born in or after 1976 (vaccinated with four doses of JE-MB vaccine or two doses of JE-CV), those born in or before 1962 (unvaccinated) and those born during 1963–1975 (vaccinated with two or three doses of JE-MB vaccine) had a 4.2-fold (95% confidence interval [CI] 3.0–5.7) and 5.9-fold (95% CI 4.3–8.1) higher risk of JE, respectively. The relative risk of working in agriculture, forestry, fishing, or animal husbandry, compared to other occupations, was 5.0 (95% CI 3.5–7.0).

### Conclusions/Significance

In Taiwan, individuals born before 1976 and those employed in agriculture, forestry, fishing, or animal husbandry had a higher risk of JE. We recommend JE vaccination for people in

**Data Availability Statement:** All relevant data are within the manuscript and its Supporting information files.

**Funding:** The study was supported by the Taiwan Centers for Disease Control, with no specific grant

funding in the public, commercial, or not-for-profit sectors. The funders had no role in study design, data collection and analysis, decision to publish, or preparation of the manuscript.

**Competing interests:** The authors have declared that no competing interests exist.

these high-risk groups who have not been fully vaccinated or have an unknown vaccination history.

## Author summary

The Japanese encephalitis (JE) virus is primarily transmitted through infected mosquito bites between animals such as pigs and birds, with humans being the dead-end hosts. JE can lead to death or long-term neurological or psychiatric complications. Vaccination is the most effective preventive measure against JE. With the implementation of comprehensive immunization programs against JE since 1960s, Taiwan has experienced a significant decline in the incidence and mortality rates of the disease. This retrospective observational cohort study utilized Taiwan's national databases from 2010 to 2022 to investigate the epidemiology of JE and identify high-risk populations, thereby to inform vaccine policy. A total of 313 JE cases were diagnosed among Taiwanese citizens with an incidence rate of 0.10 cases per 100,000 person-years and a mortality rate of 0.006 per 100,000 population per year, indicating consistently low incidence and mortality rates. Several factors were statistically significantly associated with the occurrence of JE, including male gender, individuals born before 1976, residing outside the northern region, and working in occupations related to agriculture, forestry, fishing, or animal husbandry. To prevent JE, it is recommended to offer vaccination to high-risk populations who have not completed their immunization.

## Introduction

Japanese encephalitis virus (JEV) is primarily transmitted among various animals, including pigs and birds, through the bites of infected mosquitoes, particularly *Culex tritaeniorhynchus* [1,2]. Humans, as the dead-end host, may become infected incidentally through the bites of infected mosquitoes [1,2]. Symptoms of JEV infection can vary from mild flu-like symptoms to encephalitis, the latter being characterized by high fever, headache, neck stiffness, disorientation, coma, seizures, and paralysis. About 20% of the patients with Japanese encephalitis (JE) died, and up to 50% of the survivors suffered from long-term neurological or psychiatric sequelae [3,4]. There is still no specific antiviral treatment available for JE. As JEV is transmitted in a zoonotic cycle, with humans being incidental hosts, a vaccination program would not confer herd immunity. Disease reduction depends on individual vaccination and the adoption of measures to avoid mosquito bites [5,6].

JE is a vaccine-preventable disease, and vaccination policies have successfully prevented approximately one-third of annual JE cases worldwide, leaving around 50,000 to 100,000 cases each year [5,7,8]. Vaccination has also played a crucial role in reducing JE-related deaths [3]. In Taiwan, since the 1960s, the inactivated JE mouse brain-derived (JE-MB) vaccine, derived from the JEV genotype III strain (Nakayama), has been included in the Expanded Program on Immunization for children [9]. A study conducted in Taiwan between 1971 and 2000 found that a single dose of the JE-MB vaccine can provide vaccine effectiveness of over 80% against JE for up to 30 years, and administering multiple doses can further decrease the incidence and mortality rates [10]. During the 2000s in Taiwan, the majority of JE patients were individuals who had not received the vaccine previously [5]. However, since 2010, JEV genotype I has replaced genotype III as the dominant viral strain in Taiwan [11], and the effectiveness of

JE-MB vaccine against this different strain after more than 30 years of vaccination has not been evaluated.

The risk of JE is also influenced by various other factors. Mosquitoes play a critical role in exposing humans to JEV [12]. The abundance of mosquitoes and the spread of JEV are significantly affected by multiple factors, including temperature, humidity, precipitation, altitude, land use, agricultural practices, livestock and bird densities, and socioeconomic status [13–17]. In Taiwan, the primary vector species for JEV transmission is *C. tritaeniorhynchus* [11], and the risk of JE is associated with rising temperatures and specific humidity levels [14]. The eastern region of Taiwan has reported a higher incidence of JE, but the underlying cause remains unknown [18,19]. Pig farms may pose a higher JE risk than wetlands due to the significantly higher rates of JEV infection in mosquitoes from pig farms [11]. While mosquito-borne diseases have been acknowledged as occupational hazards for individuals traveling to endemic areas [20], they may not be officially recognized as such in regions where these diseases are prevalent. Studies examining JE in specific occupations have mainly relied on questionnaire surveys [21] and descriptive epidemiology [22,23], highlighting the need for more analytical epidemiological studies.

Since there is currently no definitive method to control the vector, vaccination is considered an effective approach to preventing JEV infection [24]. The objective of this study was to explore the epidemiology of JE and identify the current high-risk groups, aiming to inform vaccination policies in Taiwan.

## Materials and methods

### Ethics statement

This study was approved by the Institutional Review Board of Centers for Disease Control, Ministry of Health and Welfare, Taiwan (No. TwCDCIRB112110). All information was anonymized and analyzed as secondary data. Informed consent was waived due to the retrospective design.

### Study population and setting

In Taiwan, JE has been categorized as a mandatory notifiable infectious disease since 1955, and in 1965, the Centers for Disease Control (CDC) established laboratory testing methods to aid in the diagnosis of JE [9,18]. Clinicians were responsible for identifying patients suspected of having JE by assessing symptoms such as fever, headache, neck stiffness, vomiting, acute psychosis (delirium, confusion, etc.), disturbance of consciousness, abnormal muscle tone, focal neurological deficit, or seizure. Subsequently, the medical staff at the hospital reported the patients' basic information and clinical characteristics to the CDC, along with samples of blood, cerebrospinal fluid, or tissue for confirmation of the diagnosis. Upon the confirmation of a case, local health authorities promptly initiated a comprehensive investigation into the affected individuals, including inquiries into their residential and occupational environments. Furthermore, information was provided to reinforce hygiene education among individuals at risk in the surrounding areas.

In order to further decrease the incidence and mortality rates of JE, Taiwan incorporated the JE vaccine into its Expanded Program on Immunization for children in 1968. Initially, children aged 3 years or younger received two doses of JE-MB vaccine [9]. By 1974, the vaccination regimen was expanded to three doses, with a booster dose of JE-MB vaccine administered one year after the initial two doses [9]. Additionally, starting in 1983, children in their first year of elementary school (around the age of six) received an extra booster dose of JE-MB vaccine [9,25]. Since 2017, JE-MB vaccine have been replaced by a live-attenuated chimeric

vaccine (JE-CV), which is administered in two doses with a 1-year interval. This vaccine implementation has effectively prevented JE and resulted in a shift of JE occurrences from children to adults. Between 1968 and 2012, the incidence rate of JE has declined significantly from 2.05 to 0.12 cases per 100,000 person-years [9,18]. Prior to 1966, over 95% of JE patients were aged 20 years or younger, whereas after 1998, over 90% were aged more than 20 years [9,26]. Moreover, the mortality rate has also been decreasing year by year [18].

## Study procedure

This retrospective observational cohort study was conducted in Taiwan utilizing nationwide databases from January 2010 to December 2022. Our investigation involved the utilization of the National Infectious Disease Reporting System (NIRDS) archive, which is maintained by the CDC. This allowed us to gather comprehensive information pertaining to laboratory-confirmed JE cases, including details such as sex, birth date, residence, environmental exposure, occupational category, symptoms, signs, date of diagnosis, and, if applicable, death date. To facilitate the identification and comparison of risk factors, we also obtained data on the sex, birth cohort, residence, and occupational category of all Taiwanese citizens throughout the duration of the study. To obtain these data, we utilized two reliable sources—the nationwide Population Data published by the Department of Household Registration, the Ministry of the Interior, and the Employment and Unemployment Statistics Enquiry System, managed by the Directorate General of Budget, Accounting, and Statistics.

## Laboratory investigations

The central laboratory of Taiwan CDC is well-equipped to conduct essential diagnostic tests for JE, encompassing serological diagnosis, molecular diagnosis, and virus isolation using cell culture. In the past, the hemagglutination inhibition assay was employed to identify a four-fold increase in plasma antibodies against both JE and dengue fever, with dengue antibody test results serving as a negative control to minimize the potential for cross-reactivity. However, a more specific diagnostic method for JE was developed by the laboratory in 1998 [27]. This method utilized an envelope- and membrane-specific capture IgM and IgG enzyme-linked immunosorbent assay (ELISA) to detect seroconversion, which subsequently became the primary approach for serological diagnosis. Starting in 2001, molecular diagnosis became another valuable tool for detecting JEV RNA in blood, cerebrospinal fluid, or tissue through real-time polymerase chain reaction [28]. Additionally, virus isolation through cell culture was another diagnostic technique employed, which involved observing characteristic cytopathic effects and detecting its antigen using an indirect immunofluorescence assay. At least one of the three diagnostic methods (serological diagnosis, molecular diagnosis, and virus isolation through cell culture) was necessary to confirm case of JEV infection.

## Data collection and definitions

Between 2010 and 2022, laboratory-confirmed JE cases were recorded monthly, from January to December. All patients and/or family members were thoroughly inquired about a range of symptoms, including fever, headache, neck stiffness, vomiting, acute psychosis, disturbance of consciousness, movement disorder, focal neurological deficit, seizures, and any other unlisted symptoms. Mortality data from the National Health Insurance Research Database, which encompasses around 99.9% of Taiwan's population, was obtained through mandatory reporting to the NIDRS. The observation period for mortality was extended to one year following the onset of symptoms or until August 1, 2023.

The laboratory-confirmed cases of JE, as well as the entire Taiwanese population, underwent a grouping based on sex at birth and birth date. Males and females were distinguished, and age categories were established to cover different age ranges: juveniles (<18 years), adults (18–64 years), and the elderly (≥65 years). Furthermore, this study classified individuals into three groups based on their birth year: those born in or before 1962, those born between 1963 and 1975, and those born in or after 1976. These groups were identified as follows: Cohort 1, consisting of unvaccinated individuals; Cohort 2, including those who received two or three doses of JE-MB vaccine; and Cohort 3, comprising those who received either four doses of JE-MB vaccine or two doses of JE-CV vaccine.

The residences were divided into four regions according to the National Development Council's classification, based on cities or counties. The northern region consisted of Taipei, New Taipei, Keelung, Taoyuan, Hsinchu, Yilan, Lianjiang, and Kinmen. The central region included Miaoli, Taichung, Nantou, Changhua, and Yunlin. The southern region encompassed Chiayi, Tainan, Kaohsiung, Pingtung, and Penghu. The eastern region comprised Hualien and Taitung (S1 Fig). Through the investigations, potential sources of exposure for JE patients, such as pig farms and pigeon coops within a one to five-kilometer radius, as well as rice paddy fields, were identified [29,30]. Individuals working in agriculture, forestry, fisheries, or animal husbandry were identified, while those employed in other industries, such as manufacturing and services, were classified as "other occupations."

### Statistical analysis

The incidence and mortality rates of JE were calculated by dividing the total number of occurrences during the follow-up period by the total person-years of follow-up. The case fatality rate was calculated by dividing the number of deaths from the disease by the total number of confirmed cases. For demographic and clinical characteristics, quantitative variables were analyzed by calculating medians and ranges, while categorical variables were examined by calculating numbers and percentages. To assess the relative risk (RR) for each category of sex, age, birth cohort, residence, and occupation, the risk of each group was compared with the risk of all the other groups. The 95% confidence interval was estimated using the Poisson distribution. Statistical analyses were performed using Statistical Analysis Software version 9.4 (SAS Institute, Cary, NC, USA).

## Results

### Incidence and mortality rates

From 2010 to 2022, a total of 313 Taiwanese citizens were diagnosed with JE through laboratory confirmation. The median annual number of JE cases was 22, ranging from 15 to 35 cases. The overall incidence rate was 0.10 cases per 100,000 person-years, which varied between 0.06 and 0.15 across the 13 study years. However, no significant change in the incidence rate was observed over the years (Fig 1A). The months of June (41%) and July (32%) had the highest occurrence of JE, followed by May (12%) and August (6%). Notably, there were no JE cases diagnosed during the months of December, January, February, March, and April (Fig 1B).

Among the 313 JE patients, fever (80%) and mental status changes (69%) were the most prevalent symptoms, followed by headaches (34%), acute psychosis (22%), and neck stiffness (18%). Less common symptoms included vomiting (10%), seizures (10%), movement disorders (9%), and focal neurological deficits (5%). A total of 17 deaths occurred within one year of diagnosis, resulting in a case fatality rate of 5.4%. The overall mortality rate was low, with an estimated rate of 0.006 per 100,000 population per year (Fig 1A). Of the 17 fatal cases, 29%

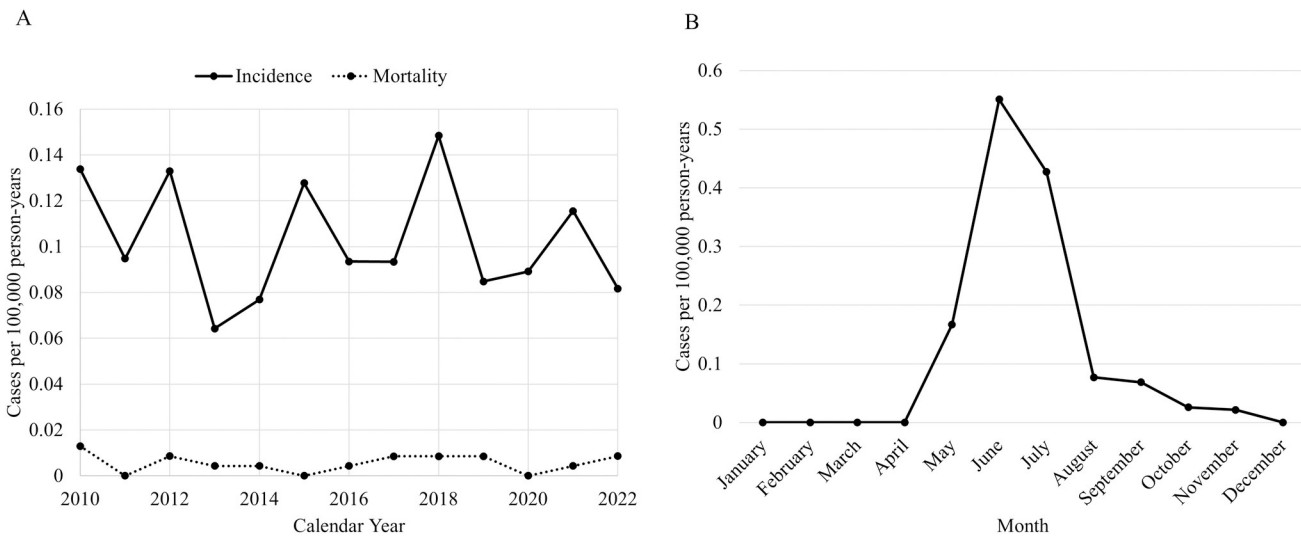

**Fig 1.** (A) The incidence and mortality rates of Japanese encephalitis ($n = 313$). (B) The incidence rate of Japanese encephalitis by month ($n = 313$). * Data is available in S1 Data file.

occurred within 30 days, while 76% and 88% occurred within 90 and 180 days of diagnosis, respectively.

## Sex, age, and birth cohort

Among the 313 patients diagnosed with JE, males accounted for 64% of the cases, with females comprising the remaining 36% (Table 1). The overall incidence rate for males (0.13 per 100,000 person-years) was higher than that for females (0.07 per 100,000 person-years) ($p < 0.0001$) (Fig 2A). The median age of the JE patients was 51 years (range, 0 to 82). Only 2%, were juveniles, while the majority were adults, making up 87% of the cases, and the remaining 11% were elderly individuals. The overall incidence rates for JE among juveniles, adults, and the elderly were 0.01, 0.13, and 0.08 per 100,000 person-years, respectively (Fig 2B). Throughout the study period, the proportions of all JE patients diagnosed in Cohorts 1, 2, and 3 were 41%, 41%, and 18%, respectively. Cohorts 1 and 2 had higher overall incidence rates of 0.15 and 0.21 cases per 100,000 person-years, respectively, while Cohort 3 had the lowest rate of 0.04 cases per 100,000 person-years (Fig 2C).

This study identified several significant factors associated with JE (Table 2). Males exhibited a higher risk compared to females (RR 1.8, 95% CI 1.4–2.3). Moreover, adults and elderly individuals were at significantly higher risk compared to juveniles (RR 11.1, 95% CI 5.0–25.0 and RR 7.3, 95% CI 3.1–17.4, respectively). Additionally, Cohorts 1 and 2 demonstrated higher risks compared to Cohort 3 (RR 4.2, 95% CI 3.0–5.7 and RR 5.9, 95% CI 4.3–8.1, respectively). Specifically, among males, the RRs for Cohorts 1 and 2 versus Cohort 3 were 3.2 (95% CI 2.2–4.8) and 5.8 (95% CI 4.0–8.3), respectively. Among females, the RRs for Cohorts 1 and 2 compared to Cohort 3 were 6.5 (95% CI 3.7–11.4) and 6.2 (95% CI 3.4–11.2), respectively.

## Residence, environmental exposure, and occupational category

The northern, central, southern, and eastern regions accounted for 22%, 28%, 44%, and 6% of the JE cases diagnosed, respectively, with corresponding incidence rates of 0.05, 0.12, 0.17, and 0.27 per 100,000 person-years (Fig 2D). The cities or counties with the highest incidence rates were Hualien in the eastern region, Nantou in the central region, Pingtung and Chiayi in the

**Table 1. The characteristics in patients with Japanese encephalitis (*n* = 313).**

| Variables | Number | Percent |
|---|---|---|
| Sex, *n* (%) | | |
| Male | 201/313 | 64% |
| Female | 112/313 | 36% |
| Age, years, *n* (%) | | |
| <18 | 6/313 | 2% |
| 18–64 | 272/313 | 87% |
| ≥65 | 35/313 | 11% |
| Birth cohort | | |
| Born ≤1962 | 129/313 | 41% |
| Born during 1963–1975 | 128/313 | 41% |
| Born ≥1976 | 56/313 | 18% |
| Residence, *n* (%) | | |
| Northern | 68/313 | 22% |
| Central | 88/313 | 28% |
| Southern | 138/313 | 44% |
| Eastern | 19/313 | 6% |
| Environmental exposure | | |
| Pig | 176/294 | 60% |
| Pigeon | 132/294 | 45% |
| Rice paddy field | 176/294 | 60% |
| Any of the above | 261/294 | 89% |
| Occupational category, *n* (%) | | |
| Students | 4/305 | 1% |
| Unemployed | 95/305 | 31% |
| Workers in agriculture, forestry, fishing or animal husbandry | 42/305 | 14% |
| Workers in other occupations [1] | 164/305 | 54% |

[1] other occupations: manufacturing industries (manufacturing, mining and quarrying, electricity and gas supply, water supply and remediation activities, construction) and service industries (wholesale and retail trade, transportation and storage, accommodation and food service activities, information and communication, financial and insurance activities, real estate activities, professional, scientific and technical activities, support service activities, public administration and defense, compulsory social security, education, human health and social work activities, arts, entertainment and recreation, other service activities).

southern region (excluding sparsely populated outlying islands) (Fig 2E). The investigations conducted by local health authorities revealed that 60% of the JE patients had exposure to pigs, 45% had exposure to pigeons, and 60% had exposure to rice paddy fields. It was observed that 89% of the JE patients had exposure to at least one of the aforementioned risk factors. Of the JE patients, 68% were employed, of whom 20% were employed in agriculture, forestry, fishing, or animal husbandry industries, and the remaining 80% were engaged in other occupations. The incidence rates among these two groups were 0.59 and 0.12 per 100,000 person-years, respectively (Fig 2F).

Residence and occupation were also identified as significant factors associated with JE. People residing in central, southern, and eastern Taiwan had higher risks compared to those living in northern Taiwan (RR 2.4, 95% CI 1.7–3.3, RR 3.4, 95% CI 2.6–4.6, and RR 5.4, 95% CI 3.3–9.0, respectively). For males, the RRs were 3.3 (95% CI 2.2–5.0), 3.1 (95% CI 2.1–4.6), and 7.0 (95% CI 3.7–13.1), respectively, while for females, the RRs were 3.3 (95% CI 2.2–5.0), 3.1 (95%

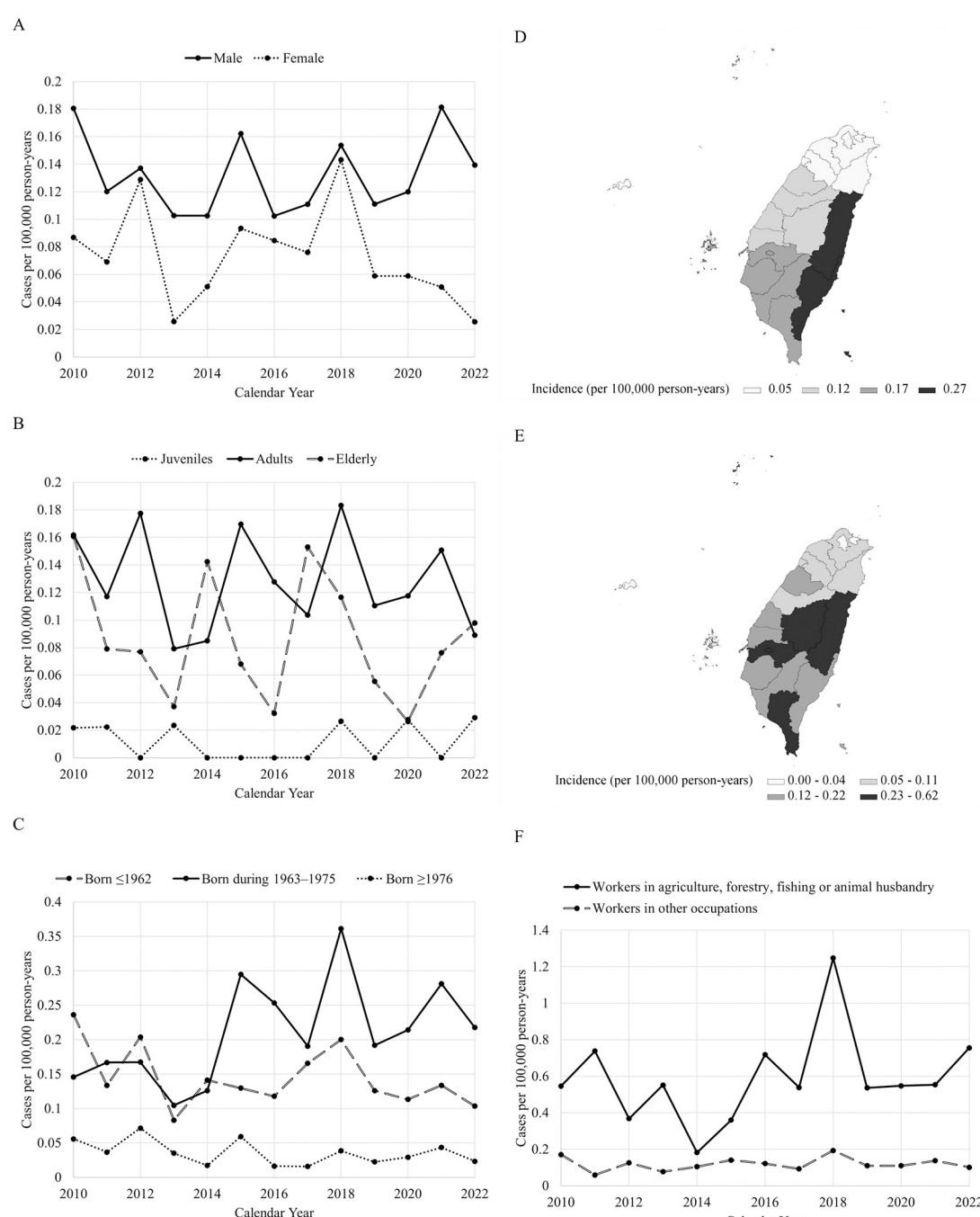

**Fig 2.** The incidence rate of Japanese encephalitis by (A) sex (*n* = 313), (B) age (*n* = 313), (C) birth year (*n* = 313), (D) region of residence (*n* = 313), (E) and city/county of residence (*n* = 313) (F) occupation (*n* = 206). * Fig 2(D) and Fig 2(E) were generated using Statistical Analysis Software version 9.4 (SAS Institute, Cary, NC, USA). The link to the base layer of the map is: https://data.gov.tw/dataset/7442. * Data is available in S1 Data file.

CI 2.1–4.6), and 7.0 (95% CI 3.7–13.1), respectively. Additionally, workers in agriculture, forestry, fishing, or animal husbandry had a higher risk compared to individuals in other occupations (RR 5.0, 95% CI 3.5–7.0). Among males, the RR was 3.7 (95% CI 2.5–5.5), and among females, the RR was 8.3 (95% CI 4.4–15.6).

**Table 2. Characteristic distributions of national population and Japanese encephalitis patients, and calculation of relative risk for disease based upon distributions by characteristics, 2010–2022.**

| Variables | National population [2] | | Confirmed Japanese encephalitis patients | | |
|---|---|---|---|---|---|
| | Number | Percent | Number | Percent | RR (95% CI) |
| Sex | | | | | |
| Male | 11,664,498 | 50% | 201 | 64% | 1.8 (1.4–2.3) |
| Female | 11,755,033 | 50% | 112 | 36% | Reference |
| Age | | | | | |
| <18 | 3,987,022 | 17% | 6 | 2% | Reference |
| 18–64 | 16,260,795 | 69% | 272 | 87% | 11.1 (5.0–25.0) |
| ≥65 | 3,175,759 | 14% | 35 | 11% | 7.3 (3.1–17.4) |
| Birth cohort | | | | | |
| Born ≤1962 | 6,766,678 | 29% | 129 | 41% | 4.2 (3.0–5.7) |
| Born during 1963–1975 | 4,723,543 | 20% | 128 | 41% | 5.9 (4.3–8.1) |
| Born ≥1976 | 12,235,486 | 52% | 56 | 18% | Reference |
| Residence | | | | | |
| Northern | 10,717,021 | 46% | 68 | 22% | Reference |
| Central | 5,790,169 | 25% | 88 | 28% | 2.4 (1.7–3.3) |
| Southern | 6,373,626 | 27% | 138 | 44% | 3.4 (2.6–4.6) |
| Eastern | 551,273 | 2% | 19 | 6% | 5.4 (3.3–9.0) |
| Occupation | | | | | |
| Agriculture, forestry, fishing or animal husbandry | 549,000 | 5% | 42 | 20% | 5.0 (3.5–7.0) |
| Other occupations [1] | 10,622,385 | 95% | 164 | 80% | Reference |

[1] Other occupations: manufacturing industries (manufacturing, mining and quarrying, electricity and gas supply, water supply and remediation activities, construction) and service industries (wholesale and retail trade, transportation and storage, accommodation and food service activities, information and communication, financial and insurance activities, real estate activities, professional, scientific and technical activities, support service activities, public administration and defense, compulsory social security, education, human health and social work activities, arts, entertainment and recreation, other service activities).

[2] National population: average mid-year population.

Abbreviations: RR, relative risk; CI, confidence interval.

## Discussion

From 2010 to 2022, Taiwan experienced consistently low incidence rate of JE and its associated mortality. However, certain factors were identified as being associated with a higher incidence of JE in Taiwan, which included male gender, individuals born before 1976, residing outside the northern region, and working in occupations related to agriculture, forestry, fishing, or animal husbandry.

Compared to other countries in the Asia-Pacific region, Taiwan has made significant strides in combating JE [31]. Similar to South Korea and Japan, Taiwan has successfully reduced the occurrence and death rates of JE [18,31–33]. Various factors have contributed to this achievement, including urbanization, reduced rice paddy field sizes, relocation of pig farms, and the use of pesticides to control mosquito vectors. However, the most crucial factor has been the implementation of comprehensive immunization programs against JE. Initially introduced in the 1960s, the program administered two or three doses of the JE-MB vaccine, resulting in a vaccination rate of 80% or higher. Subsequently, the policy was updated to administration of four doses, with an outstanding vaccination rate of 95%. As a result, Taiwan has maintained a stable JE incidence rate of 0.10 cases per 100,000 person-years over the past three decades [9], with a record-low death rate of 0.006 per 100,000 population per year in the past decade [18]. The success of the JE immunization program has also highlighted the crucial role of

vaccination in reducing fatalities [3], as evidenced by Taiwan's one-year case fatality rate of 5.4%, which was lower than those in the previous records [18]. However, it is important to acknowledge the potential for a slight increase in delayed deaths resulting from long-term neurological or psychiatric sequelae [4].

In Taiwan, different numbers of vaccine doses may result in varying levels of protection. Cohort 1, with a higher likelihood of mosquito exposure during childhood, primarily acquired JEV infection naturally. On the other hand, Cohorts 2 and 3 obtained antibodies through various vaccination programs implemented following environmental changes. During the follow-up period, Cohorts 1 and 2 exhibited significantly higher incidence rates of JE compared to Cohort 3. This could be attributed to lower antibody levels in Cohorts 1 and 2, unlike nearly all people of Cohort 3 who had undergone complete vaccination against JE and might continue to have higher antibody levels [9]. Furthermore, Cohort 2 had a higher incidence rate than Cohort 1, potentially due to incomplete vaccination that might fail to provide the same long-lasting immunity as natural infection. Another factor to consider was the vaccines developed for JEV genotype III might have reduced effectiveness against JEV genotype I [34]. Overall, this study suggested that individuals who had not received four doses of JE-MB vaccine exhibited significantly higher incidence rates than those who had received four doses of JE-MB vaccine or two doses of JE-CV. Our findings may provide additional insights into previously limited information on vaccine effectiveness at the population level [4,19].

There were several other factors that contributed to the occurrence of JE in Taiwan. Males were found to have a higher risk than females, likely due to their involvement in outdoor activities. Nationwide data between 2010 and 2022 revealed that 56% of those engaged in agricultural, forestry, fishing, and animal husbandry work were males, while 44% were females. Females in Cohort 1 had a higher relative risk compared to Cohort 3, possibly due to their lesser exposure to early natural infections and development of antibodies. This observation could also potentially explain the higher incidence rate of infections in males, likely because they have more exposure to JEV [35]. JE was predominantly prevalent during June and July, influenced not only by the favorable temperature and humidity for mosquito growth but also by pig breeding [14,18]. The agriculture, forestry, fishing, and animal husbandry industries posed a higher risk of disease transmission, likely due to increased exposure to pigs, birds, and farmland. However, in Taiwan, an endemic area for JEV infection, criteria for determining whether JEV infections in individuals residing in such areas are qualified as occupational diseases have not been established. This study revealed the link between JE and occupational factors by identifying a notable number of JE patients in Taiwan over the past 13 years who were involved in specific occupations.

The incidence of JE in Taiwan exhibited regional variations, with the eastern region having the highest incidence, followed by the southern and central regions, while the northern region had the lowest incidence. Previous research has indicated that the seropositive rates of Cohort 2, sampled in eastern, southern, central, and northern Taiwan, were 32%, 39%, 40%, and 44%, respectively. These findings suggested a lower vaccination coverage during the early stages of the vaccination program in certain region. Moreover, the distribution of industries varied across different regions, with 80% of agriculture and animal husbandry workers concentrated in the central, southern, and eastern regions. In these regions, the proportion of such workers was 5–15 times higher than in the northern region, with the eastern region having the highest proportion. As a result, individuals residing in the eastern region of Taiwan were likely at a higher risk of contracting JEV infection due to a lower vaccine coverage and a greater proportion of agriculture and animal husbandry workers.

The production of JE-MB vaccine was discontinued due to safety concerns and the requirement for multiple doses, despite its immunogenicity. In contrast, JE-CV has been shown to be

safe, highly immunogenic, and capable of providing long-lasting immunity. Notably, a single dose of JE-CV is sufficient to confer protective immunity, comparable to the effect of three doses of JE-MB vaccine in adults [36]. In light of the unavailability of JE-MB vaccine, JE-CV could be considered as a replacement or used as a booster or catch-up regimen [26]. However, it is important to note that, similar to JE-MB vaccine, JE-CV is derived from the JEV genotype III strain (SA14-14-2). While some partial cross-protection between genotype I and III strains has been observed in mice, the effectiveness of JE-CV against the genotype I strain was lower than its effectiveness against the genotype III strain, as demonstrated in both mice and swine [34,37]. Further analysis of vaccine effectiveness in humans would be necessary, and the development of a bivalent vaccine could be a potential alternative strategy for vaccination against JE.

## Limitations

There were several limitations to this study. First, the incidence and mortality of JE may have been underestimated as the awareness of JE and decision to notify suspected cases were at the discretion of attending physicians. However, since notification is mandatory, and Taiwan CDC is the only laboratory in Taiwan with capability to confirm the diagnosis of JE, the likelihood of missing to notify suspected cases was considered low. Second, there might be inaccuracies in the established databases. We did not have vaccination records for all individuals, and our evaluation of vaccine effectiveness relied solely on the risk of the birth cohort. Furthermore, the scope of investigation by local health authorities regarding exposure to pigs, pigeons, and rice paddy fields might not have been extensive enough given the long-distance migration ability of *C. tritaeniorhynchus* [38]. To address this, we assessed the occupational risk of agriculture, forestry, fishing, and animal husbandry workers to evaluate the exposure risk. Third, it is important to note that factors identified as high risk for JE do not necessarily prove causation. Possible risk factors were not simultaneously evaluated in one model, and future studies should consider adjusting for confounding factors if the databases can be integrated. Finally, it should be emphasized that the prevalent JEV strains in Taiwan after 2010 were genotype I. Therefore, the findings of this study may not be generalizable to other countries, and policies should be adjusted based on research results relevant to the specific region or country.

## Conclusions

JE is a life-threatening disease that should not be underestimated, despite its low incidence and mortality rates in Taiwan. It poses a particular risk to individuals born before 1976 and those employed in agriculture, forestry, fishing, or animal husbandry. To prevent JE, vaccination should be offered to high-risk populations who have not received complete immunization.

## Supporting information

**S1 Fig. The distribution map of cities and counties in Taiwan.** S1 Fig was generated using Statistical Analysis Software version 9.4 (SAS Institute, Cary, NC, USA). The link to the base layer of the map is: https://data.gov.tw/dataset/7442.
(TIF)

**S1 Data. The number and incidence of Japanese encephalitis by month, sex, age, birth year, region of residence, city or county of residence, and occupation.**
(XLSX)

## Acknowledgments

The authors would like to thank all staff in the Taiwan government agencies for the investigations and data preservation.

## Author Contributions

**Conceptualization:** Jen-Yu Hsu, Wan-Chin Chen.

**Data curation:** Jen-Yu Hsu.

**Formal analysis:** Jen-Yu Hsu.

**Funding acquisition:** Jen-Yu Hsu.

**Investigation:** Jen-Yu Hsu.

**Methodology:** Jen-Yu Hsu, Wan-Chin Chen.

**Project administration:** Jen-Yu Hsu.

**Resources:** Jen-Yu Hsu.

**Software:** Jen-Yu Hsu.

**Supervision:** Jen-Yu Hsu, Chien-Ching Hung, Tsung-Pei Tsou, Wan-Chin Chen.

**Validation:** Jen-Yu Hsu, Chien-Ching Hung, Tsung-Pei Tsou.

**Visualization:** Jen-Yu Hsu, Chien-Ching Hung.

**Writing – original draft:** Jen-Yu Hsu.

**Writing – review & editing:** Jen-Yu Hsu, Chien-Ching Hung.

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
