## [Decision Letter · Decision Letter 0]

30 Jul 2023

Dear Dr. Hsu,

Thank you very much for submitting your manuscript "Epidemiology and risk factors of Japanese encephalitis in Taiwan, 2010–2022" for consideration at PLOS Neglected Tropical Diseases. As with all papers reviewed by the journal, your manuscript was reviewed by members of the editorial board and by several independent reviewers. The reviewers appreciated the attention to an important topic. Based on the reviews, we are likely to accept this manuscript for publication, providing that you modify the manuscript according to the review recommendations. 

Sincerely,

Daniel M Parker

Academic Editor

Victoria Brookes

Section Editor

Reviewer's Responses to Questions

**Key Review Criteria Required for Acceptance?**

**Methods**

-Are the objectives of the study clearly articulated with a clear testable hypothesis stated?

-Is the study design appropriate to address the stated objectives?

-Is the population clearly described and appropriate for the hypothesis being tested?

-Is the sample size sufficient to ensure adequate power to address the hypothesis being tested?

-Were correct statistical analysis used to support conclusions?

-Are there concerns about ethical or regulatory requirements being met?

Reviewer #1: Clear descriptions of the aims, demographic variables, and datasets, as well as the diagnostic methods considered for confirming a case of JEV infection were provided.

The stratification used to calculate relative risks is useful, but without any sort of confounding analysis.

Reviewer #2: This is a very well-written manuscript about Japanese encephalitis (JE) in Taiwan.

The objectives were clearly specified. It's a 13-year retrospective epidemiological study from 2010 to 2022 that documented 313 cases of JE. 

Several risk factors were studied, including gender, age, date of birth, immunization protocols, region and city of residence, and occupation.

The method for calculating the fatality rate must be verified.

**Results**

-Does the analysis presented match the analysis plan?

-Are the results clearly and completely presented?

-Are the figures (Tables, Images) of sufficient quality for clarity?

Reviewer #1: Overall, the result section is well-developed; although it may lead to some confusion to only have the risk factor analysis at the end of the section, when the comparisons are made earlier on. The figures and the tables are suitable.

Reviewer #2: (No Response)

**Conclusions**

-Are the conclusions supported by the data presented?

-Are the limitations of analysis clearly described?

-Do the authors discuss how these data can be helpful to advance our understanding of the topic under study?

-Is public health relevance addressed?

Reviewer #1: The conclusions drawn from the data are well-supported, and the paper effectively addresses most of its limitations. It provides a thorough explanation of the potential reasons behind the observed differences in risk groups. A notable limitation arises from solely stratifying the data based on one variable of interest, such as sex or profession, without implementing a risk factor analysis to adjust relative risks (RRs) for potential confounding factors (by multiple stratification or regression).

Reviewer #2: The authors suggest a new vaccination protocol that could improve the level of immunization of Taiwanese population at risk of JE.

**Editorial and Data Presentation Modifications?**

Reviewer #1: 1) The paper could benefit of controlling for confounding, perhaps through multiple stratification, building upon the stratification performed in the paper, to simultaneously adjust for several of the aforementioned risk factors. For instance, the higher risk of JE in males may be attributed to their involvement in high-risk professions, increased participation in outdoor activities (as discussed in the paper's discussion section), lower vaccination coverage... Otherwise, it could be mentioned in the discussion some literature in Taiwan that conducted similar analysis with confounding.

2) Consider clarifying that “herd immunity may not necessarily work” (line 81) due to the zoonotic nature of the virus (so immunizing humans would not eliminate the virus from the region). Citation: https://www.ncbi.nlm.nih.gov/books/NBK219063/

3) In the abstract, should “population” in line 43 be “population per year” or “person-years”?

4) In line 92, it is mentioned that the majority of Japanese encephalitis patients in Taiwan during the 2000s were unvaccinated individuals. Additionally, in lines 276-278, it is stated that the vaccination rate has been high since the 1960s. Hence, when discussing the younger population (born after 1962), as the recommendation of the paper is to provide vaccination, it could be beneficial to mention some existing literature and available data on the reasons behind vaccination rates not reaching 100% in Taiwan and where research should move to. This exploration may shed light on potential factors contributing to higher incidence rates in men, beyond differences in outdoor activities mentioned in lines 300-1 (but without mentioning other studies with similar results). 

5) Is there any available information regarding the vaccination status of individuals in Cohorts 2 and 3, considering that vaccination rates were high but not ~100%? Only in the abstract it is mentioned that vaccination persons "without vaccination" or "unknown vaccination status", but the lack of data on individual vaccination status of cases could perhaps be best clarified as a limitation.

Reviewer #2: (No Response)

**Summary and General Comments**

Reviewer #1: The article by Hsu et al. is clear and the writing style is fluid and easily understandable. The article sheds light on an important regional topic, aiming to enhance the understanding of the epidemiology of Japanese encephalitis. It delves into various aspects including the clinical manifestations, high-risk groups, vaccine history/policy and seasonality of JE in Taiwan.

Reviewer #2: (No Response)

PLOS authors have the option to publish the peer review history of their article (what does this mean?). If published, this will include your full peer review and any attached files.

Reviewer #1: No

Reviewer #2: No

Figure Files:

Data Requirements:

Reproducibility:

References

---

## [Decision Letter · Decision Letter 1]

24 Sep 2023

Dear Dr. Hsu,

We are pleased to inform you that your manuscript 'Epidemiology and risk factors of Japanese encephalitis in Taiwan, 2010–2022' has been provisionally accepted for publication in PLOS Neglected Tropical Diseases.

Before your manuscript can be formally accepted, please address the final comments from reviewers. You will also need to complete some formatting changes, which you will receive in a follow up email. A member of our team will be in touch with a set of requests.

Best regards,

Daniel M Parker

Academic Editor

Victoria Brookes

Section Editor

Please address the final comments from the reviewers.

Reviewer's Responses to Questions

**Key Review Criteria Required for Acceptance?**

**Methods**

-Are the objectives of the study clearly articulated with a clear testable hypothesis stated?

-Is the study design appropriate to address the stated objectives?

-Is the population clearly described and appropriate for the hypothesis being tested?

-Is the sample size sufficient to ensure adequate power to address the hypothesis being tested?

-Were correct statistical analysis used to support conclusions?

-Are there concerns about ethical or regulatory requirements being met?

Reviewer #1: The authors have adequately addressed the comments from the reviewers. I do have a minor suggestion for further refinement:

On lines 333-334, could you provide the 95% CI for the overall incidence rates for both males and females? If this is not feasible, offering a p-value might suffice to conclusively state that male rates are higher, even though this conclusion can be suspected from Figure 2A.

Reviewer #2: This is a very well-written manuscript about Japanese encephalitis (JE) in Taiwan.

The objectives were clearly specified. The study population and diagnostic methods are well described.

**Results**

-Does the analysis presented match the analysis plan?

-Are the results clearly and completely presented?

-Are the figures (Tables, Images) of sufficient quality for clarity?

Reviewer #1: (No Response)

Reviewer #2: The result section is well developped. The "sex, age and birth cohort" section has been improved as well as the "Residence, environmental exposure, and occupational category" section. Figures and tables are adequate.

**Conclusions**

-Are the conclusions supported by the data presented?

-Are the limitations of analysis clearly described?

-Do the authors discuss how these data can be helpful to advance our understanding of the topic under study?

-Is public health relevance addressed?

Reviewer #1: (No Response)

Reviewer #2: The findings from the data are clearly supported.

The discussion has been improved. The study-Limitations were addressed and discussed.

The authors suggest a new vaccination protocol that could improve the level of immunization of Taiwanese population at risk of JE.

**Editorial and Data Presentation Modifications?**

Reviewer #1: (No Response)

Reviewer #2: In section 'Please add funding details as follow up to "financial disclosure": replace "commericial" by commercial.

Line 347: Replace "was" by were in "...awareness and decision ...suspected cases was at the discretion...".

**Summary and General Comments**

Reviewer #1: (No Response)

Reviewer #2: This is a very well-written manuscript about Japanese encephalitis (JE) in Taiwan. . It's a 13-year retrospective epidemiological study from 2010 to 2022 that documented 313 cases of JE.

Several risk factors were studied, including gender, age, date of birth, immunization protocols, region and city of residence, and occupation.

The authors suggest a new vaccination protocol that could improve the level of immunization of Taiwanese population at risk of JE.

PLOS authors have the option to publish the peer review history of their article (what does this mean?). If published, this will include your full peer review and any attached files.

Reviewer #1: **Yes: **Luís-Jorge Amaral

Reviewer #2: No

---

## [Editor Report · Acceptance letter]

27 Sep 2023

Dear Dr. Hsu,

We are delighted to inform you that your manuscript, "Epidemiology and risk factors of Japanese encephalitis in Taiwan, 2010–2022," has been formally accepted for publication in PLOS Neglected Tropical Diseases.

Best regards,

Shaden Kamhawi

co-Editor-in-Chief

Paul Brindley

co-Editor-in-Chief
